# Nanotechnology-Based Approaches for Voriconazole Delivery Applied to Invasive Fungal Infections

**DOI:** 10.3390/pharmaceutics15010266

**Published:** 2023-01-12

**Authors:** Laís de Almeida Campos, Margani Taise Fin, Kelvin Sousa Santos, Marcos William de Lima Gualque, Ana Karla Lima Freire Cabral, Najeh Maissar Khalil, Ana Marisa Fusco-Almeida, Rubiana Mara Mainardes, Maria José Soares Mendes-Giannini

**Affiliations:** 1Pharmaceutical Nanotechnology Laboratory, Department of Pharmacy, Midwest State University (UNICENTRO), Alameda Élio Antonio Dalla Vecchia St, 838, Guarapuava 85040-167, PR, Brazil; 2Department of Clinical Analysis, School of Pharmaceutical Sciences, São Paulo State University (UNESP), Rodovia Araraquara Jaú, Km 01, Araraquara 14801-902, SP, Brazil

**Keywords:** antifungals, sustained drug release, fungal infections, nanoparticles

## Abstract

Invasive fungal infections increase mortality and morbidity rates worldwide. The treatment of these infections is still limited due to the low bioavailability and toxicity, requiring therapeutic monitoring, especially in the most severe cases. Voriconazole is an azole widely used to treat invasive aspergillosis, other hyaline molds, many dematiaceous molds, *Candida* spp., including those resistant to fluconazole, and for infections caused by endemic mycoses, in addition to those that occur in the central nervous system. However, despite its broad activity, using voriconazole has limitations related to its non-linear pharmacokinetics, leading to supratherapeutic doses and increased toxicity according to individual polymorphisms during its metabolism. In this sense, nanotechnology-based drug delivery systems have successfully improved the physicochemical and biological aspects of different classes of drugs, including antifungals. In this review, we highlighted recent work that has applied nanotechnology to deliver voriconazole. These systems allowed increased permeation and deposition of voriconazole in target tissues from a controlled and sustained release in different routes of administration such as ocular, pulmonary, oral, topical, and parenteral. Thus, nanotechnology application aiming to delivery voriconazole becomes a more effective and safer therapeutic alternative in the treatment of fungal infections.

## 1. Introduction

Fungal infections are a growing threat to global public health. Most of these fungal infections are superficial, but some species can cause life-threatening illnesses. Immunocompromised patients are at higher risk for fungal infections, including organ transplant, oncology, HIV/AIDS, and, more recently, SARS-CoV-2 patients [1,2,3,4,5]. It has also occurred in immunocompetent patients as a secondary infection [6,7]. Systemic fungal infections usually originate either in the lungs, after conidia inhalation (*Aspergillus*, *Cryptococcus*), or from endogenous microbiota (*Candida* spp.) as a result of infected lines or leakage from the gastrointestinal tract, and may spread to many other organs. These pathogens, under certain circumstances, can spread, causing fatal infections responsible for more than one million deaths worldwide each year. Thus, the systemic spread of fungi is a critical step in developing these deadly infections. These infections are associated with high morbidity and mortality rates, especially in some hospital settings [8,9,10,11]. If appropriate therapy is delayed, systemic fungal infections are medical emergencies with high mortality rates.

The distribution of mycoses varies according to several factors, including the region and the epidemiological conditions, and may be classified as superficial [12], cutaneous [13,14,15], subcutaneous [16], and systemic [17]. Among the most common fungal infections prevails in order of importance infections by *Cryptococcus* sp. [18], *Candida* sp. [19], and *Aspergillus* sp. [20]. These infections are cited in a document published this year as belonging to the critical priority group, these based not only on prevalence or incidence, but also considering multicriteria decision analysis [21].

Treatment for these infections involves four main antifungal classes targeting different fungal cell structures [22]. However, studies have shown a reduction in the susceptibility of fungi compared to the available classes [23,24]. The main factors related to decreased antifungal susceptibility are drug target overexpression, efflux pumps, and amino acid substitution [25,26]. In addition, another limitation to the use of current antifungals is the presence of adverse effects, the main ones being nephrotoxicity and hepatotoxicity [27,28].

Voriconazole (VCZ) is an antifungal of the azole class, resulting from a structural modification of fluconazole with a broad spectrum of activity, and is commonly used for prophylaxis and treatment of invasive fungal infections [29,30,31,32,33]. VCZ requires therapeutic monitoring and maintenance dosages for a prolonged period due to the recurrence of infection [34,35,36]. The main limitation of VCZ therapy is the inter-individual variation of its plasma levels due to factors such as liver function, polymorphisms in cytochrome P450 isoenzymes [37,38], drug interactions, liver disease, and cancer [39,40,41]. Toxicity occurs when serum levels are in the supratherapeutic range, especially in prolonged treatments, including phototoxicity, hallucinations, hyponatremia, and others [42,43]. VCZ has been the second alternative because it has less favorable pharmacological properties. To overcome these features is necessary to develop a new approach as nanostructured systems that could be excellent carriers for antifungal drugs, reducing toxicity and targeting their action. The application of nanostructured systems for antifungal therapy began in the 1990s with the development of lipid formulations of amphotericin B [44].

Nanoparticles have been studied for antifungal therapy, and results show evident improvements in drug aspects such as solubility and stability in water, increased bioavailability, and tissue penetration, which result in increased efficacy and reduced toxicity [45,46,47]. Moreover, due to the prolonged drug release profile, nanoparticles can maintain drug plasma levels balanced in the therapeutic range, which is especially important for antifungals with a low therapeutic index or that present non-linear pharmacokinetics, such as VCZ [44]. Also, drug-loaded nanoparticles can improve the fungal inhibition profile even in lower concentrations compared to plain antifungals [48,49].

Therefore, this review presents the main nanotechnology-based systems developed for delivering VCZ and discusses their effectiveness as a new avenue for treating fungal infections.

## 2. Fungal Infections

### 2.1. Pulmonary Aspergillosis

Pulmonary aspergillosis is caused by the conidial saprophytic fungus *Aspergillus*, found in the soil and the air, dust from civil construction, and medical dispositive, which affects immunocompromised patients or those with pre-existing lung disease [20,50]. The main pathologies related to pulmonary aspergillosis are allergic bronchopulmonary aspergillosis (ABPA), chronic pulmonary aspergillosis (CPA), and invasive pulmonary aspergillosis (IPA), the determinant of these infections being the interactive relationship between the fungus and the host [51].

Currently, 446 species of *Aspergillus* are described in the literature; however, only 20 are related to fungal infections [52,53,54]. The main species that cause invasive and pulmonary infections are *A. fumigatus*, *A. flavus*, *A. niger*, *A. terreus*, *A. nidulans*, *A. calidoustus*, *A. sydowii*, and *A. versicolor* [51]. Studies compared *Aspergillus* species isolated from respiratory samples of CPA patients with those from colonization. The species that were most often isolated from patients were *A. fumigatus* (48.3%), *A. niger* (29.2%), *A. flavus* (8.3%), *A. terreus* (2.5%), and *A. nidulans* (0.8%) [55].

#### 2.1.1. Allergic Bronchopulmonary Aspergillosis (ABPA)

ABPA is characterized by inflammation in the lung due to hypersensitivity to the fungus *Aspergillus* sp. It is related to patients with asthma and cystic fibrosis, some patients with tuberculosis [56], or who suffer from chronic obstructive pulmonary disease (COPD) [57].

The most common symptoms are chronic cough, wheezing, and recurrent pulmonary infiltrates, with a chest X-ray showing dilatation of the bronchi with phlegm (bronchiectasis). However, patients with ABPA may present nonspecific symptoms that are like other pre-existing pulmonary pathologies, reporting symptoms such as productive cough, wheezing, fever, chest pain, sweating, weight loss, blood expectoration (hemoptysis), and golden-brown mucus secretion is characteristic of this pathology [56].

The diagnosis is based on three criteria: predisposition due to the presence of asthma or cystic fibrosis; the mandatory criterion, a positive skin test for *Aspergillus* or elevated IgE against *A. fumigatus* (total IgE > 1000 UI/mL); and finally, at least two of the three support criteria—recent eosinophil count in patients without the use of corticosteroids (>500 cell/L), radiographic characteristics of ABPA how the bronchiectasis, and serum precipitins or IgG against *A. fumigatus* [58,59].

Treating asymptomatic patients with controlled asthma has no clear benefits; however, careful monitoring of patients is necessary to define treatment. The main point of treating ABPA is suppressing the hyperimmune response and reducing the mycological load, using glucocorticoids and antifungals, such as itraconazole, as the first-choice treatment. If the pathological picture does not improve, the second-choice treatment involves using VCZ orally or posaconazole [60]. Biologic therapies have recently shown promising results with omalizumab and mepolizumab [61,62].

#### 2.1.2. Chronic Pulmonary Aspergillosis (CPA)

CPA is a chronic progressive pulmonary disease commonly caused by *Aspergillus fumigatus* and affects immunocompromised patients and those with pre-existing lung disease [63,64]. The most usual form among most patients is chronic cavitary pulmonary aspergillosis (CCPA) [65]; however, if treatment does not occur or is inadequate, it can progress to chronic fibrosing pulmonary aspergillosis (CFPA) [20]. Conditions that predispose to CPA are ABPA, COPD, lung cancer, asthma, pneumonia, and fibrocavitary sarcoidosis [55].

The most frequent symptoms are chronic productive cough, weight loss, fever [66], dyspnea [55], hemoptysis with nodules, cavities, and fungi balls [67]. The persistence of symptoms for more than three months helps its diagnosis [68].

Diagnostic criteria are related to the consistent appearance in clinical and radiological aspects, an immune response to *Aspergillus* sp., being culture positive for *Aspergillus* sp. for sputum or bronchoscopy samples and/or detectable galactomannan and/or positive *Aspergillus* sp. DNA by PCR polymerase chain reaction [63,69,70].

The VCZ treatment is the first choice, followed by posaconazole, which demonstrates fewer side effects [71]. Antifungal treatment intravenously can be used in the event of failure with previous therapies, but its response is slow, and its use is indicated for at least six months [72].

#### 2.1.3. Invasive Aspergillosis (IA)

Invasive aspergillosis includes invasive pulmonary aspergillosis (IPA), *Aspergillus* sinusitis, disseminated aspergillosis, and several types of the single organ [70]. Invasive aspergillosis (IA) in hematology/oncology patients presents as a primary or rupture infection, which can become refractory to antifungal treatment and has a high associated mortality. Other risk groups of emerging patients include patients in intensive care with severe respiratory viral infections, including COVID-19 [73]. The size of the conidia, thermotolerance, hydrophobins, and melanin on the conidial surface, adaptability to the host environment, and angioinvasive nature all contribute to pathogenicity [74]. The global prevalence of aspergillosis reaches an estimated 3,000,000 cases per year of chronic pulmonary aspergillosis and 300,000 cases per year of IA [75].

IPA is the most severe evolutionary form of this pathology characterized by the growth of *Aspergillus* sp. hyphae in the lung, and it affects immunocompromised patients with a high mortality rate [76]. Tracheobronchitis is an infection limited to the tracheobronchial region and is an exclusive form of IPA [77,78]. Invasive rhinosinusitis is also recognized as another form of pathology [59]. The progression of the IPA is rapid, from days to weeks, with a mortality rate between 30% and 85%, with *A. fumigatus* being the most frequent cause of IPA [76]. There are several risk factors related to IPA, such as prolonged profound neutropenia, human immunodeficiency virus (HIV) infection, acute leukemia, hematopoietic stem cell transplantation, and diabetes mellitus, among others [20,79].

Acute symptoms can cause intravascular thrombosis and pulmonary hemorrhagic infarction [80]. Clinical manifestations that are more common are cough, fever, chest or pleuritic pain, dyspnea, and hemoptysis [77]. This form may be confused with bacterial pneumonia [8].

Invasive pulmonary aspergillosis is an opportunistic mycosis, challenging to diagnose due to environmental *Aspergillus*. Current recommendations suggest that epidemiological, radio-clinical, and biological data support the diagnosis of aspergillosis, as well as early computed tomography (CT) scans to identify the two main features, angioinvasive and invasive airway aspergillosis. Although CT findings are not entirely specific, they usually allow for early initiation of therapy before mycological confirmation of the diagnosis. Confirmation is based on microscopy and culture of respiratory specimens, histopathology in case of biopsy, and, most importantly, detection of *Aspergillus* galactomannan using an immunoassay in serum and bronchoalveolar lavage. Histology allows proving the diagnosis of aspergillosis, but biopsy is not always possible in immunosuppressed patients.

New antifungal agents have been developed in the last two decades: new azoles (VCZ, posaconazole, and isavuconazole), lipid formulations of amphotericin B (liposomal amphotericin B, and amphotericin B lipid complex), echinocandins (caspofungin, micafungin, and anidulafungin). Thus, medical imaging and serum galactomannan antigen currently form the basis of the screening approach, although both have some limitations in specificity.

The first-choice treatment for IPA is VCZ, which should be administered once the diagnosis is confirmed to reduce patient mortality [81]. The therapy is initially administered intravenously until the patient shows improvement and then substitutes for oral therapy. Alternative therapies in IPA treatment are liposomal amphotericin B, isavuconazole, itraconazole, and echinocandins, which can be used as a combination therapy with other antifungals. The duration of treatment with an antifungal is around six to twelve weeks, but it can take months for up to over a year [80]. For the treatment of invasive aspergillosis (IA), the most recently used therapy has been isavuconazole, showing a high efficacy in the treatment of aspergillosis in immunocompromised patients, a lower potential for Drug–Drug Interactions (DDIs), and no risk of QT prolongation (heart rhythm disturbance) that can cause a rapid and chaotic heartbeat [82,83,84].

### 2.2. Candida Infections

Several species can cause *Candida* infections, but the most common is *Candida albicans*. Among the most common clinical manifestations of *Candida* sp. are invasive candidiasis, oral candidiasis, denture stomatitis, and candidemia neonatal [85,86,87]. Invasive candidiasis is one of the most aggressive forms of this disease, and affects immunocompromised patients, those who have undergone some organ transplant or are undergoing chemotherapy treatment, with a mortality rate above 70% in the latter group [88,89,90]. *Candida* spp. are common commensal organisms in the skin and gut microbiota, and disruptions in the cutaneous and gastrointestinal barriers (for example, owing to gastrointestinal perforation) promote invasive disease [90].

There are more than 15 species of *Candida* sp. capable of causing infections in humans; however, the five most common species that cause infections are *C. albicans*, *C. glabrata*, *C. tropicalis*, *C. parapsilosis*, and *C. krusei*, where *C. albicans* is responsible for 40 to 60% of cases worldwide [91,92,93,94]. Another species that has drawn attention is *C. auris*. Infections and outbreaks caused by this species in hospital settings have recently increased. Difficulty in its identification, multidrug resistance properties, the evolution of virulence factors, high associated mortality rates in patients, and long-term survival on surfaces in the environment make this species particularly problematic in clinical settings [95,96,97]. Associated with the increase in COVID-19 infections, a trend toward bacterial, fungal, and viral superinfection has been observed. An important co-infection agent is *C. auris* due to its multidrug-resistant nature and easy transmissibility. Patients with comorbidities, immunosuppressive states, and intubated and mechanically ventilated patients are more likely to contract the fungal infection; therefore, being placed in the critical group of human pathogenic fungi by the WHO [21,98,99].

Risk factors for candidemia are using the venous catheter, admission to the intensive care unit, broad-spectrum antibiotics, abdomen surgery, parenteral nutrition, neutropenia, acute renal failure, malignancy, and burns [11].

Candidemia symptoms are related to the risk factors predisposing to this type of infection. However, clinical manifestations such as blood infections related to catheter use, septic shock, and eye involvement are observed [92].

The diagnosis is based on the detection of *Candida* sp. by the culture method, which in turn has low sensitivity, and other tests are frequently requested, such as the detection of antigen (1,3)-β-D-glucan, tube antibodies *C. albicans*, and techniques based on molecular biology such as PCR, RT-PCR and MALDI-TOF MS [100,101,102].

The treatment protocols use four different classes of drugs: polyenes, triazoles, echinocandins, and flucytosine [103,104]. The first-choice medicine for treating candidiasis and invasive *Candida* infections is echinocandins; however, other protocols are used and consider the species causing the infection and the associated clinical conditions [105]. Monotherapy protocols are amphotericin B, in the traditional or liposomal form, anidulafungin, caspofungin, micafungin, fluconazole, and VCZ. However, the combined therapies of amphotericin B and fluconazole or amphotericin B and flucytosine also apply [106,107].

### 2.3. Cryptococcosis

Cryptococcosis is a systemic fungal infection caused by *Cryptococcus* spp., the main species are *Cryptococcus neoformans*, and *Cryptococcus gattii* can affect both immunocompetent and immunosuppressed patients in the pulmonary, extrapulmonary, and disseminated form through the Central Nervous System (CNS), causing meningoencephalitis and rarer cases of cutaneous or transplant-associated mycosis [108]. Cryptococcal meningitis has become an infection of global importance reaching 1 million new infections per year; despite both species having many characteristics in common, there are some differences regarding geographic distribution, environmental niches, host predilection, and clinical manifestations that should be emphasized [109,110].

Cryptococcosis was recently highlighted by the world health organization as the first highest-priority microorganism, surpassing even *C. auris*, more recently described and of worldwide concern due to its resistance. Inserting new therapies for mycoses, especially those referred to here, is essential in this context, according to the number of deaths from HIV associated cryptococcol meningitis, an estimated 181,000 cases worldwide represent 15% of all AIDS-related deaths [1,21].

The pathogenic species of *Cryptococcus* sp. are divided into five different serotypes subdivided into different molecular types described in Table 1.

*Cryptococcus* spp. virulence is associated with the polysaccharide capsule, which increases pathogenicity, modulates the immune response, and protects against oxidative stress [114,115,116]. The presence of melanin in the cell wall protects against antifungal drug attacks at elevated temperatures and contributes to the modulation of the immune response [117,118]. Furthermore, the secretion of enzymes, such as urease, has played a significant role in fungal metabolic pathways [119,120].

The transmission occurs from inhaling desiccated airborne yeast cells, or possibly sexually produced basidiospores, into the lungs from the feces of pigeons in the environment or different tree species [121]. Risk factors for *Cryptococcus* spp. infection are HIV, rheumatic diseases, solid organ transplantation, corticosteroid or immunosuppressive therapies, chronic decompensated liver disease, diabetes mellitus, sarcoidosis, kidney problems, use of monoclonal antibodies, lymphoid diseases, and overexpression syndrome IgE and IgM [1,110,122].

Patients with pulmonary infection caused by *Cryptococcus* spp. are usually asymptomatic and account for a third of the number of cases, and the diagnosis is made by the presence of nodules on radiographic examination [113]. Some patients also have symptoms like pneumonia and can progress quickly to acute respiratory distress syndrome even without CSN involvement [110]. According to authors [113], the most common symptoms are fever and dry cough, but the patient may have dyspnea, chest pain or discomfort, malaise, or no symptoms.

There are some options to diagnose *Cryptococcus* sp, isolation of the fungus in biological fluids using culture and identification, histopathology, serological detection of the cryptococcal capsular polysaccharide antigen (CrAg), and molecular methods [122].

Treatment of patients with cryptococcosis considers risk factors or other pathologies that affect the patient concurrently being categorized by the host’s immune status, organ involvement, and respiratory cryptococcosis with or without CNS problems [1]. The treatment is based on using antifungal agents or drainage of cerebrospinal fluid and surgical resection [123].

The first-choice treatment of cryptococcosis is using Amphotericin B, despite the reported hepatotoxic and nephrotoxic effects [124]. Following the use of fluconazole, mainly because it has a lower cost, some studies indicate a reduction in susceptibility to this class of antifungals [125]. In addition, *Cryptococcus* sp. has intrinsic resistance to echinocandins [126]. In this sense, the treatment of cryptococcosis is a challenge, and the search for new alternatives has grown in recent years [127], such as the combination of antifungal drugs [128], the use of nanotechnology, and the search for new molecular targets [129,130,131].

## 3. Voriconazole

### 3.1. General Aspects

VCZ (Vfend^®^. by Pfizer^®^, via tablet and IV) is an antifungal agent belonging to the azole class, derived from a structural modification of fluconazole and approved by the Food and Drug Administration (FDA) in 2002 [132,133]. It has low solubility in water (0.098 mg/mL), log P 1.82, and pKa 2.01 and 12.7 [134]. VCZ acts by inhibiting cytochrome P450 (CYP 450)-dependent 14α-lanosterol demethylation, an important step in cell membrane ergosterol synthesis by fungi [135,136]. It is fungicidal against most molds, except Mucorales. It is active against all species of *Aspergillus*, including *A. terreus*, which is generally resistant to amphotericin B, hyaline fungi, including *Fusarium* spp. and members of the *Scedosporium apiospermum* complex, except *S. prolificans* [137,138,139]. VCZ is fungistatic, like all azoles, against *Candida* spp. Species that are inherently resistant to fluconazole, such as *C. krusei*, are susceptible to VCZ and some *C. glabrata* that are resistant to fluconazole are susceptible to VCZ, but many strains develop resistance to fluconazole also become resistant to VCZ [140,141]. *Candida albicans* was the predominant species, causing up to two-thirds of all cases of invasive candidiasis. However, a change to *Candida* spp. non-albicans, such as *C. glabrata* and *C. krusei*, with reduced susceptibility to commonly used antifungals, have been recently observed. On the other hand, *Candida auris*, an emerging pathogen, is highly tolerant to azoles and often resistant to several drugs [142,143]. VCZ shows good in vitro activity against other yeasts, including *Cryptococcus neoformans*, *Trichosporon asahii*, and *Saccharomyces cerevisiae* [144,145]. Finally, VCZ has activity against *Blastomyces dermatitidis*, *Coccidioides* spp., *Histoplasma capsulatum* and *Paracoccidioides brasiliensis*, but it is not active against *Sporothrix schenckii* [146,147,148].

Although it is still widely used in the treatment of invasive aspergillosis, the consequences of the non-linear pharmacokinetics of VCZ are one of its main limitations [53,149]. Since dosages have wide interpersonal variation, they can reach subtherapeutic or toxicity levels [150,151]. Antifungals of the same class, such as posaconazole and isavuconazole, have been used as an alternative in risk groups for treatment with VCZ [152,153].

### 3.2. Pharmacokinetics

VCZ is a class II drug with low solubility and high permeability [154,155,156]. Low water solubility makes it difficult to administer by different routes of administration [157]. VCZ is commercially available in tablets, oral suspension, and intravenous solutions [158,159,160]. An essential point of the VCZ pharmacokinetics is that they are non-linear, presenting wide inter-individual variability according to cytochrome P450 polymorphisms [161].

#### 3.2.1. Absorption

VCZ is rapidly absorbed, and its plasma levels depend on body weight, age, route of administration, presence of polymorphisms, and inflammation, among other factors [162,163,164,165]. After oral administration, the maximum plasma concentration occurs within 1 to 2 h, reaching about 96% [166]. However, a wide variation of absorption (35% to 83%) appears in clinical studies associated with the high or low activity of the CYP2C19 enzyme [167,168,169].

#### 3.2.2. Distribution

The estimated volume of distribution is between 2 to 4.6 L/kg, suggesting intravascular and extravascular distribution [41]. Binding to plasma proteins has not been defined yet, despite efforts to measure it through in vitro and in vivo studies. The binding proteins reported are albumin and glycogen-α-1-acid [170,171].

#### 3.2.3. Metabolism

VCZ is extensively metabolized in the liver by enzymes of cytochrome P450 (CYP450), comprising CYP2C19, CYP3A, and CYP2C9 subfamilies [38], which may present genetic polymorphism and their expression is affected, resulting in a phenotype of poor, intermediate, or rapid metabolizers [172]. CYP2C19 enzymes convert VCZ into its inactive metabolite, VCZ-N-oxide [173].

Factors such as pre-existing diseases and the presence of CYP450 polymorphisms are responsible for the non-linear metabolism of the VCZ, resulting in dose-dependent auto-inhibition and saturation of metabolism [161,174,175,176]. In addition, polymorphisms can lead to unwanted interactions with other drugs [177].

#### 3.2.4. Excretion

Only 2% VCZ is excreted unchanged in the urine, with an elimination half-life of approximately six hours [178,179].

### 3.3. Toxicity and Drug Monitoring Therapeutics

Therapeutic monitoring has been suggested in clinical practice to reduce adverse effects and increase the therapeutic efficacy and safety of VCZ [180,181]. Its unpredictable nonlinear pharmacokinetics, with wide variation in serum levels between individuals, makes therapeutic regimens difficult due to the polymorphisms associated with its primary metabolism, drug interactions, and oral bioavailability [135,182].

An investigation conducted by Zonios with 95 patients at a medical center, about 7 to 18% of patients treated with VCZ experienced adverse effects, including hallucinations, visual disturbances, photosensitivity, and hepatotoxicity [183]. Among the adverse effects in patients with prolonged therapy, hepatotoxicity and neurotoxicity were frequent [42,184,185]. In another study carried out by Epaulard, phototoxicity was present in 8% of patients treated in 61 case reports [186].

Therapeutic monitoring of VCZ levels has been investigated due to the effects of its non-linear pharmacokinetics, especially in children [187], the elderly [37], patients in Intensive Care Units (ICU), and for hematological and inflammatory diseases [188,189]. The central monitoring measures have been the evaluation of CYP2C19 polymorphisms [190] and plasma drug concentration [189,191], in addition to the evaluation of liver function and other measures [192,193]. Monitoring VCZ plasma concentrations is carried out with well-standardized Liquid Chromatography coupled with Mass Spectrometry and High-Performance Liquid Chromatography methods [194,195].

It has been observed an increase in clinical efficacy, reduction in adverse effects [196,197,198], and reduction in intra and inter-individual variability [199] with dose monitoring, being a safe alternative for using VCZ in the treatment of invasive fungal infections [200].

## 4. Nanotechnology-Based Voriconazole Delivery Systems

The interest in studying nanostructured systems has been maintained over the years due to their numerous advantages for developing of science and society. In pharmaceutical and biomedical fields, the application of nanoparticles has modified the way drug transporting in the body [201]. Hydrophilic or hydrophobic molecules, with highly variable molecular weight, labile (proteins, nucleic acids, vaccines) or not, can be carried by nanostructures and safely transported to exert their pharmacological activities [202,203]. The modification of several drug parameters has already been reported after their nanoencapsulation, such as the improvement in solubility [204], controlled release, improvement in pharmacokinetics [205], protection against degradation [206,207], and targeting to specific tissues [208]. Furthermore, these systems can be used as diagnostic [209] and therapeutic tools [210].

Nanoparticles are mainly classified according to their composition, and depending on the method used to obtain them, different supramolecular structures are obtained. A variety of materials has been used to obtain organic nanoparticles, such as polymers (natural or synthetic), lipids, surfactants, or proteins (animal or vegetable) [211]. These materials require biocompatibility, biodegradability, and specific mechanical and/or thermal properties. Based on these parameters, structures such as nanocapsules, nanospheres, liposomes, solid lipid nanoparticles, nanostructured lipid carriers, nano and microemulsions, cyclodextrins, and others can be developed [212,213]. The different supramolecular arrangements give these nanostructured systems differences in size, shape, drug loading capacity, drug release profile, biological half-life, interaction with cells and biodistribution [214]. The material’s chemical composition confers different electrical charge characteristics to the surface of the particles, which interferes with its physical and biological stability and interaction with cells and tissue internalization. The surface of nanoparticles can be chemically modified by interaction and cellular targeting [215].

Considering all limitations regarding the use of VCZ, nanotechnology-based delivery systems represent an excellent approach. Furthermore, the loading of VCZ in nanoparticles, in addition to improving the physicochemical and biological aspects of the drug in the conventional routes of administration (oral and parenteral), could propose alternative routes (topical, intranasal, pulmonary, ocular, vaginal, and others), as can be represented in Figure 1. Table 2 lists some studies of nanoparticle formulations containing VCZ. We briefly present the composition of formulations, preparation methods, route of administration, and the in vitro and in vivo results.

### 4.1. Lipid Nanoparticles

Lipid nanocarriers are potential drug delivery systems because they allow a controlled and specific target release [250,251]. The main lipid nanocarriers are the liposomes, solid lipid nanoparticles (SLNs), and nanostructured lipid carriers (NLCs) [252,253].

Liposomes are spherical vesicles formed by one or more phospholipid bilayers, surrounded by an aqueous compartment. Hydrophilic drugs can be loaded in the aqueous core while hydrophobic drugs are trapped into the lipid bilayer [254].

SLNs consist of carriers composed of a solid lipid core surrounded by a surfactant layer. The lipid is presented as a solid at room temperature and is smelt at higher temperatures. Hydrophobic drugs are better loaded in SLNs than hydrophilic ones. This first generation of SLNs presented limitations such as low physical stability, limited encapsulation efficiency, and expulsion of the drug during storage. It occurs due to the formation of a perfectly crystallized lipid matrix after solidification of the lipid. The second generation of SLNs are the NLCs, the core of which is constituted by a solid lipid and a liquid lipid (oil), forming a disordered lipid arrangement, avoiding the drawbacks of SLNs [255].

Among the solid lipids commonly used for SLNs are those with a low melting point and solidity at body temperature, in addition to surfactants and co-surfactants [256]. The development of SLN has improved the encapsulation efficiency and stability of drugs using a mixture of liquid, solid lipids, and surfactants [257,258].

SLNs plays a significant role in improving pharmacokinetic characteristics and reducing the adverse effects of antifungals [259]. Amphotericin B was the first antifungal to be commercialized in colloidal dispersion systems, lipid complexes, or liposomes to improve drug solubility and reduce its adverse effects, [45,260].

The ability to improve drug solubility through lipid formulations has also been explored in VCZ delivery systems, to topical [160,232] and ocular administration [157]. In the study carried out by Andrade, an NLS was developed with 75% encapsulation efficiency that allowed safe release into ocular tissue in ex vivo tests [232].

In a study by Liu et al. (2023), a liposome containing voriconazole showed a greater capacity for binding to the chitin of the fungal cell wall of *C. albicans*. Furthermore, the in vivo study improved the delivery efficacy of voriconazole [261].

In vivo study of vaginal infections by candida albicans showed a significant reduction in infection after 48 and 72 h compared to free voriconazole, indicating sustained release of the drug [220].

In general, lipid nanoparticles containing voriconazole have been developed mainly for ocular delivery and this is justified by the compatibility and permeation capacity of these systems in this tissue, favoring a safe delivery of drugs. Studies with SLNs have shown increased permeation, mainly ocular, and allowed controlled and sustained release, increasing the therapeutic efficacy of these systems for VCZ delivery [46,243,244].

### 4.2. Polymeric Nanoparticles

Among the administration systems and targeted delivery of drugs and molecules, polymeric nanoparticles play a prominent role in this scenario. The main feature is the ease of interacting with pharmacokinetic parameters in administering these systems, such as absorption, bioavailability, and excretion [262,263].

There are two types of biodegradable polymers used, with synthetic polymers such as Poly (lactide) (PLA) [264], Poly (lactide-co-glycolic) (PLGA) [265], Poly (ε-caprolactone) (PCL) [266], and natural polymers such as chitosan [267], zein [268], casein [269], alginate [270], gelatin [271], and albumin [272].

The methods for obtaining nanostructured systems with polymeric nanoparticles are based on two main techniques, dispersion and polymerization [273]. However, these systems still present some challenges as drug delivery systems: the non-scalable procurement methods, the safety and cost of development, and limitations as a system, such as overcoming some biological and stability barriers [274].

In this sense, several systems have been studied in different pathophysiological conditions to overcome these challenges. Among the main diseases that have already been approved and have effective results is cancer, in systems that release doxorubicin and paclitaxel, for example [275,276,277]. Studies also use neurodegenerative [278] and cardiovascular diseases [279,280].

Despite the challenges, polymeric nanoparticles are promising for their use and development advantages [281]. They provide the encapsulated molecule with protection against degradation in some routes of administration, such as oral administration [282]. They enable the controlled release of the target-specific drug and surface modification with the incorporation of ligands [283,284].

Some nanoparticles have also been explored for VCZ delivery. Chitosan-coated PLGA nanoparticles were developed by Paul (2018) using the solvent evaporation emulsification method to improve the bioavailability and release of VCZ through the pulmonary route. The study obtained nanoparticles with 68.57% encapsulation efficiency and a sustained and slow release in vitro and in vivo, which allowed the retention of VCZ in the lung and plasma [230].

In another study by Rençber and Karavana (2018), chitosan nanoparticles were developed to carry VCZ by the emulsification method for topical administration. The nanoparticles had almost 99% encapsulation efficiency. The release was characterized by diffusion, and the method was considered promising for topical administration. The authors justify the result by the association of chitosan, which has mucoadhesive properties [229].

Das (2015) was able to increase the bioavailability of VCZ via the pulmonary route in PLGA nanoparticles developed by solvent evaporation emulsification, with an average diameter of approximately 300 nm. The system allowed increased pulmonary deposition and prolonged release, thus improving the pharmacokinetic characteristics of VCZ [248]. Another study with PLGA was developed by Sinha, Mukherjee and Pattnaik (2013), and the system allowed more significant pulmonary deposition and prolonged release of VCZ [285].

The characteristic of allowing the incorporation of different molecules into the polymeric nanoparticle allows for an increase in the interaction of the nanostructured system with the target tissue [229,283,284]. This has been explored in the literature with chitosan, which is known to have a mucoadhesive property and allows the increase in the delivery of VCZ in the target tissue [218,221,230,247].

### 4.3. Protein Nanocarriers

Protein nanoparticles are also described as polymeric nanoparticles, as they are classified as natural biopolymers. The main proteins used in developing these systems are proteins of animal origin, such as albumin, gelatin, collagen, and fibroin, and proteins of vegetable origin, such as zein and gliadin [286]. The procurement of nanoparticles occurs through different techniques, such as emulsification, desolvation, complexation by coacervation, electrospray deposition, and nanoprecipitation [287,288]. The protein nanoparticles can be classified, according to composition, into several types: solid spherical nanoparticles, plate-shaped nanoparticles, and nanogels [289]. Its applications involve different areas such as drug delivery, nutrient and metabolite delivery, gene delivery, tissue engineering, photodynamic therapy, growth factor delivery, vaccines, and cosmetics [288,290].

Human serum albumin nanoparticles containing VCZ were successfully developed by nab^TM^ technology. The aim of the investigation was to optimize critical process parameters such as the homogenization pressure, the number of homogenization cycles, and the organic solvent. After determining the critical parameters, the authors obtained nanoparticles in the range of 35 to 85 nm, whose size was adequate for the proposed parenteral administration, with an encapsulation efficiency of around 69%. After 6 h, the system had already released 85% of the encapsulated VCZ, which made this system promising for further studies [231].

In addition, nanoparticles for delivering voriconazole are promising, although their application varies according to the need for the route of administration. It is possible to see that lipid formulations seem to improve delivery by the ocular and topical route and protein and polymeric formulations were explored for oral and pulmonary delivery [217,222,234,237,240]. The physicochemical characteristics of nanoparticles are important for the treatment of fungal infections, as they allow a better interaction between the fungus and the nanostructure.

In general, using nanotechnology for voriconazole encapsulation and release is a strategy to increase therapeutic activity since studies have reported increased tissue permeation with reduced toxicity, mainly due to the observed sustained and controlled release [211,216]. In addition, it is possible to observe an improvement in the solubility and bioavailability of VCZ [230,237].

Despite this, there is still a need for more comprehensive studies regarding the cytotoxicity and antifungal potential of these nanoparticles, mainly in in vivo studies.

## 5. Other Systems for Voriconazole Delivery

### 5.1. Cubosomes

Cubosomes are reversed bicontinuous cubic phase liquid crystalline nanoparticles capable of transporting and releasing drugs at the specific target site and have a high rate of encapsulation efficiency. These characteristics are due to the high surface area between the hydrophilic and hydrophobic regions present in its structure. They are composed of a lipid and surfactant/stabilizer, which extends in three dimensions and two nanochannels interwoven, but not connected [291,292,293].

Recently, cubosomes have been studied to increase and control the bioavailability of VCZ after ocular application. VCZ suspensions showed a low ocular residence time, which leads to an increase in the frequency of applications, while VCZ-containing cubosomes demonstrated a prolonged release profile compared to the suspension. Cubosomes were also coated with chitosan, increasing transcorneal permeation and residence time at the site [224].

### 5.2. Cyclodextrins

Cyclodextrins (CD) are cyclic oligosaccharides composed of six or more glucose units formed from an enzymatic reaction on starch. The most important naturally occurring forms are the a-, b-, and g- CD, whose structural shape resembles a cone, where the inner cavity has hydrophobic characteristics and the outer part is hydrophilic. Among the advantages of using DC in nanotechnology is its ability to improve drug solubility and organoleptic properties, stability, and safety [294,295].

CDs were used to improve VCZ solubility, dissolution rate, and chemical stability through complexation in an aqueous solution, followed by spray-dryer drying [294]. Another study carried out the complexation of CD and VCZ to produce a thermosensitive gel based on Poloxamer 407 and Poloxamer 188 for vaginal application. VCZ formulation complexed as CD showed greater uptake of the drug by the vaginal tissue compared to the formulation that used dispersed VCZ [236].

One of the most recent studies involves CD to improve the solubility of VCZ and the production of hydrogels for ophthalmic application since there is no formulation for this purpose. The authors compared the residence time and ophthalmic safety of b- and g- CD complexes compared to the control (VFEND). The results demonstrated good corneal permeability of VCZ, longer residence time, and ophthalmic safety [296]. Therefore, the use of DC to improve the performance of VCZ has shown promise.

## 6. Clinical Trials

Currently, on the Clinical Trials website, there are 189 records of studies with VCZ, of which only two registered studies involve the evaluation of nanocarrier formulations. The PHASE 2 study under registration NCT04110860 conducted by the Minia University of Egypt evaluated the application of a nanoemulsion-based gel containing VCZ in patients with tinea versicolor once or twice a day, compared to a placebo (nanoemulsion-based gel without VCZ). The study included 30 volunteer patients aged between 10 and 60 years, both sexes, with tinea versicolor, excluding pregnant women, nursing mothers, and immunocompromised patients. Patients’ clinical improvement, satisfaction, and duration of treatment were evaluated. Adverse effects were recorded, photographs of the lesions were taken before the start of treatment, during and at the end, and the clinical improvement criteria used by the physicians used a quartile rating scale. No results from this study have been published on the Clinical Trials website [297].

The PHASE 1 study under registration NCT01657201 conducted by Samyang Biopharmaceuticals Corporation of South Korea evaluated intravenous administration of VCZ-loaded-PNP 200 mg versus Vfend 200 mg in 59 healthy male patients aged 20 to 45 years to assess pharmacokinetic parameters and safety. The study was randomized, crossover, and an open intervention model; the pharmacokinetic parameters evaluated during the 24 h of the study were AUClast, Cmax, AUCinf, Tmax, T1/2, and CL [298]. Through this brief research, we can conclude that studies involving the application of nanostructured systems containing VCZ are still limited and that the studies carried out are still in the initial stages of research.

## 7. Future Prospects

It was observed that advances in the production of nanostructures containing voriconazole with a focus on topical, ocular, and pulmonary administration are significant in relation to other drug application routes. Despite the diversity of developed systems, there are still limitations regarding efficacy and safety studies. Most of the studies compiled in this work employ in vitro tests to demonstrate its performance, while in vivo tests are rarely explored. The reduced applicability of in vivo tests and the lack of use of alternative models, both used for toxicity and pharmacokinetic assessments, indicate the need for efforts in this area. Another point is the development of new drugs using nanotechnology, where clinical studies are rare, which indicates that there is still much to be done for research to proceed and benefit the population. In this sense, the application of nanotechnology is encouraged for better performance of fungal drugs, which allow a higher efficacy and safety for the treatment of different fungal infections that explore the interaction of nanoparticles with yeast and filamentous fungi, improving the prospective application of these nanoparticles.

## 8. Conclusions

The therapeutic management of opportunistic fungal infections such as aspergillosis, candidiasis, and cryptococcosis is still a challenge, which is associated with pharmacokinetic and toxicity limitations of currently available antifungals. Among them, VCZ has a wide sub- and supratherapeutic variation due to its metabolism and CYP450 polymorphisms, which reduce its therapeutic efficacy and amplify its adverse effects.

Nanocarrier systems loaded-VCZ offer numerous advantages. Among the results demonstrated, we can highlight the improvement in solubility and dissolution rate of VCZ in an aqueous medium. The release control presented different release profiles, most of them allowing a prolonged or sustained release in the presence of an initial burst effect. In addition, pharmacokinetic parameters improve the increase in mucosal adhesion and safety for topical applications. The physicochemical characteristics change according to the nanostructured system, and these data are verified according to widely disseminated techniques for characterization. Nanotechnology applied to the use of VCZ has the potential to improve parameters related to drug targeting, release, and therapeutic effects for overcoming the biological and physicochemical hurdles.

## Figures and Tables

**Figure 1 pharmaceutics-15-00266-f001:**
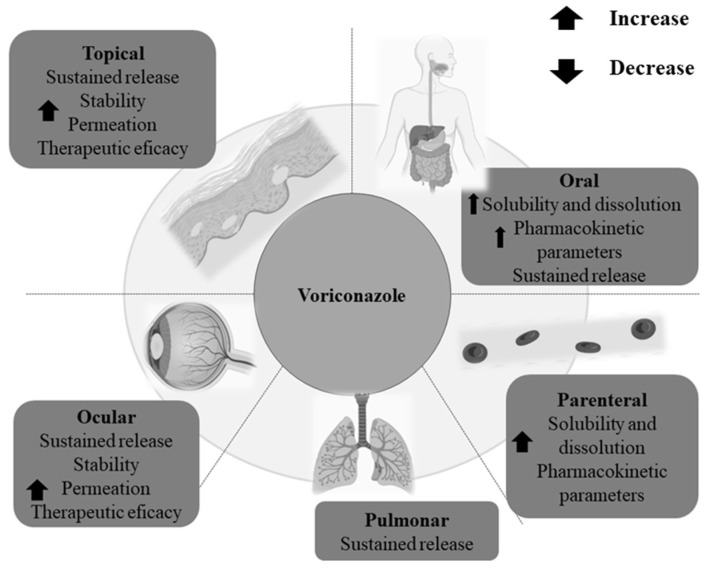
Possible routes for voriconazole delivery from nanoparticles.

**Table 1 pharmaceutics-15-00266-t001:** Classification of *Cryptococcus* species.

Species and Varieties	Serotype	Molecular Types
*C. neoformans* var. *grubii* ^1^	A	VN I, VN II
*C. neoformans* var. *neoformans*	D	VN IV
*C. neoformans*	AD	VN III
*C. gattii*	B	VG I, VG II, VG III, VG IV
*C. gattii*	C	VG I, VG II, VG III, VG IV

^1^ Variety responsible for most cases worldwide. Adapted from [110,111,112,113].

**Table 2 pharmaceutics-15-00266-t002:** Some aspects of studies on nanoformulations containing voriconazole in recent years.

Composition	Method of Preparation	Route of Administration	Size	Toxicity	In Vitro	In Vivo	Reference
Chitosan, Sodium Lauryl Sulfate, Poloxamer, Benzalkonium Chloride	O/W Solvent Emulsification Technique	Ocular	219.3 nm	A study on egg chorioallantoic membrane indicated nanoparticle is not irritating	In *C. albicans* there was a reduction in MIC compared to free VCZ; NPs loaded in situ gel had MIC at 0.06 µg/mL over 1 µg/mL of free VCZ. Mucoadhesion was increased with nanoparticles and prolonged release for up to 8 h. An Ex vivo study revealed increased permeation of the VRC from the nanoparticles in the cornea.		[216]
Chitosan, sodium lauryl sulfate, propylene glycol, Polyethylene glycol-4000	Spray Dryer	Topical	160–500 nm		There was greater retention in the skin and low retention in the stratum corneum. The nanoparticles showed even greater inhibitory activity on *C. albicans* than on free VCZ, with an inhibition halo of 17.55 mm for the nanoparticle and 9.25 mm for free VCZ.		[217]
Chitosan, Sodium Tripolyphosphate (TPP), dipalmitoylphosphatidylcholine (DPPC)	Ionic gelation	Pulmonary	228–255 nm	Cell viability and uptake studies showed cytocompatibility in A549 and Calu-3 lung epithelial cells.	Higher efficacy in *Candida* sp. and *Aspergillus* sp. about free VCZ in laboratory strains and equal inhibition in clinical isolates.	Pharmacokinetic study showed an increase of almost 5, 4, and 3 times in the area under the curve, T_max,_ and residence time in the lungs.	[218]
Eudragit RS 100 PVP (Polyvinylpyrrolidone) PVA (Polyvynil Alcohol)	Quasi-emulsion solvent evaporation	Ophthalmic	138 nm		The system showed a higher inhibitory effect on *C. albicans* at lower concentrations than VCZ injection.	Increased corneal permeability in rats	[219]
Glyceryl monooleate Glyceryl monostearate Maisine	High Shear Homogenization and Ultrasonication	Vaginal	322.72 nm	Normal morphological features on histopathology, with results similar to the negative control	The in vitro release profile compared the aqueous suspension of VCZ and the optimized formulation and verified a sustained release profile, where 70% was released after 8 h.	The reduction of fungal load in Wistar rats was higher when using the nanoparticle about the VCZ suspension.	[220]
Chitosan, Tween 80, Sodium tripolyphosphate	Ionic gelation	Topic	199–232 nm	A skin irritation study was performed in albino rabbits, and no signs of skin irritation and inflammatory cell infiltration were observed.	The release profile was slower in PBS medium pH 7.4 compared to pH 5.5, releasing 82% VCZ in 24 h at the most acidic pH. The ex vivo test using mouse skin (mice) demonstrated that a limited amount of VCZ permeated in the receptor medium, higher for the film with suspended VCZ than the film with VCZ-polymeric nanoparticles. The latter presented higher deposition (15.05%) in the stratum corneum concerning the other formulations and the deposition of 54.76% in the epidermis and dermis. In the antifungal test, the film based on VCZ-PNPs showed the highest retention zone against *Candida* sp. (20 mm) and *Aspergillus* sp. (17 mm).	The histological study confirmed its safety, which makes it suitable for topical application.	[221]
Surfactant, cyclodextrin	Micellization thermodynamic	Topic	13–16 nm		A 24-h dialysis membrane release test was performed with different formulations and the release varied from 72 to 75%. The model that best explained the release was the first-order model (r^2^ = 0.99).		[222]
Sodium taurodeoxycholate	Emulsification	Topic	21–24 nm		The Franz cell release test had performed for 10 h, and the amount released corresponded to 30% of the formulation’s VCZ. The kinetic model that best explains it was Higuchi’s (r^2^ = 0.9842). The permeation test was performed on hulls for 24 h, and no amount of VCZ was detected in the receiving medium. The concentration found in the hull increase 10×.		[223]
Monoolein, Pluronic F127, chitosan	Cubosomes by Melt dispersion emulsification	Ocular	109–243 nm		The release test had performed in a dialysis bag for 24 h. The cubosomes showed a biphasic release profile, with a burst effect at 30 min followed by a sustained release for 24 h.	Pharmacokinetic was performed in albino rabbits by ocular instillation. The results showed that the chitosan-coated cubosomes showed higher C_max_ than the VCZ-suspension (4.44 and 3.52 ng/mL, respectively; *p* < 0.0001). A similar performance had obtained for parameters AUC_0–8_ and AUC_0-∞_.	[224]
Isopropyl myristate, PEG 400, Tween 80^®^, Span 80^®^.	Self-nano emulsifying	Ocular	21 nm		Ex vivo: nanostructured VCZ formulation demonstrated an increase in VCZ transcorneal permeation compared to the commercial formulation.		[225]
IPM, PEG 400, Tween 80^®^, Span 80^®^.	Self-nano emulsifying (SNEDDS)	Ocular	21 nm	Eye irritation and damage tests were performed on rabbits. The findings demonstrate that the VCZ nanoformulation was well tolerated and capable of ocular delivery.	When comparing VCZ marketed formulation and VCZ SNEDDS the antifungal activity (MIC) showed similar results for *Candida* sp. and significantly lower MIC (*p* < 0.001) for *A. fumigatus*.	The pharmacokinetic evaluation was superior to the commercial one, presenting the following results: AUC_0–8 h_: 16,200 µg/mL; T_max_: 2 h; C_max_: 5577 µg/mL.	[226]
Polaxâmero 188, Dodecilsulfato de sódio, cloreto de cetiltrimetilamônio	Nanospray dryer	Oral	421 nm	Subacute treatment with API-NP up to a concentration 80 mg/kg of body weight did not cause adverse toxicological effects in the organs evaluated.	Improved solubility, dissolution, and release of VCZ in the aqueous medium.	API-NP showed improved in pharmacokinetic parameters (AUC, C_max_) compared to API tablets and VFEND^®^. Increased bioavailability, sustained release, and less inter-individual variability.	[227]
Kolliphor^®^ HS 15, Sulfobutyl ether-β-Cyclodextrin	Self-assembly method		13–15 nm				[228]
L-α-Fosfatidilcolina, Polissorbato 80, Witepsol W35, Ácido esteárico e Compritol 888 ATO	High-pressure homogenization	Ophthalmic	182 nm		The dialysis release test demonstrated that the SLNs were able to release VCZ. The formulation demonstrated antifungal efficacy against *Aspergillus flavus* and *Candida glabrata.*		[157]
Chitosan, EUD RS 100	Spontaneous Emulsification	Topic	217 nm		Desired physicochemical characteristics of the formulation for administration in mucosa showing mucoadhesion and release mechanism by dialysis and constant diffusion in vitro.		[229]
Chitosan, PLGA, PVA	Multiple emulsion by solvent evaporation	Pulmonary	154–277 nm	After six hours, a greater accumulation of VCZ was detected in the liver than in the urine, suggesting that urinary clearance decreases.	Both formulations, PLGA-NP, coated or not with chitosan, showed sustained VCZ release after 24 h and followed the Korsmeyer–Peppas kinetic model.	Both formulations showed uniform distribution in the alveoli and sustained release up to 72 h, about free VCZ. VCZ levels were detected in the lung and plasma after administration. The C_max_ was achieved earlier by the chitosan-coated formulation, but the time required was the same when compared.	[230]
Albumin	nab^TM^-technology	Parenteral	81 nm		Up to 2× increase in VCZ solubility.		[231]
NLC (Tween 80, capric caprylic triglycerides, Span 85, cetylpyridinium chloride (CPC), Compritol 888 ATO)	Microemulsion	Ocular	250 nm	The weak irritant in HET-CAM irritation test.	Therapeutic delivery after 30 min in ex vivo permeability assessment.		[232]
SBE-β-CD	Electrospinning	Parenteral	*		It demonstrated ease in promoting rapid dissolution for reconstitution of the pharmaceutical form.		[233]
Mannitol (MAN)	Thin Film Freezing (TFF)	Pulmonary	3 µm		The VCZ nanoaggregates 95:5 formulation showed better in vitro performance in the aerosol performance test, FPF 73.6%, and in dissolution test.		[234]
Compritol 888 ATO, Miglyol 812N, Gelucire 44/14, Solutol HS 15 e Tween 80.	Melt High-Pressure Homogenization		45 nm		There was no difference in MIC for VCZ–NLC and free VCZ. However, at low concentrations, the inhibition rate of planktonic cells of *C. albicans* was higher for VCZ–NLC; there was also a reduction in the biofilm cell density. There is an increase in the efficiency of the VCZ.		[235]
HP-β-CD + P407, P188	Spray drying	Vaginal	*		The addition of mucoadhesive polymers increases mucoadhesion and sustained drug release of VCZ.	VCZ uptake was higher when administered in the VCZ HP-β-CD formulation, whose Cmax was 7.13 µg/g at two h post-dose, an increase 3.4× greater than HP-β-CD or VCZ in dispersion.	[236]
Mannitol, TBA (*tert*-Butyl alcohol)	Spray freezes drying	Pulmonary	2–4 µm		All test formulations showed complete dissolution within the first 5 min.	V8 (intratracheal) had a higher concentration of VCZ in the lungs when compared to VFEND^®^(IV). After 30 min, the concentration of VCZ was lower in the liver and spleen, and there was no significant difference in the kidneys.	[237]
Solid Lipid (Compritol 888 ATO or Stearic Acid), Span 80/60, Tween 80	High-shear homogenization followed by probe ultrasonication		286 nm		VCZ-SLN reduced the MIC50 value for all the tested *Aspergillus fumigates* (susceptible and resistant) about free VCZ.		[238]
Phosphatidylcholine, cholesterol, α-tocopherol	Lipid-film hydration followed by extrusion	Intravenous	95 nm	Accumulation of VCZ in the liver and kidneys was lower in the liposomal form.	*Candida* sp. was more susceptible than strains of *Aspergillus* sp., but there was no difference between VCZ liposome and VFEND^®^.	There was a difference in the pharmacokinetic parameters for liposome and VFEND^®^. Liposome reduced the deputation by half; the AUC_0–24_ increased 2.5× and reduced the volume of distribution of the VCZ.	[239]
Phosphatidylcholine (Liposomes) Compritol, Miglyol, Tween 80 e Span 85 (NLC)	Film hydration followed by extrusion (Liposomes) and microemulsion (NLC)	Topic	114 nm		The percentage of VCZ release was higher for NLC (40%) than for liposome (15%) after six hours. The gel formulation showed significant accumulation only with liposomes. Liposome deposition in the follicle produces the greatest amount in the stratum corneum. While NLC has a faster and deeper release. The MIC50 to *Trichophyton rubrum* result was similar for both formulations.		[240]
Tween 80, ethanol and oleic acid.	Microemulsion	Topic	10 nm		In the pork skin permeation test, the accumulation of VCZ in the stratum corneum and the rest of the skin was higher for the microemulsion concerning the commercial formulation. The antifungal activity was better for the microemulsion containing VCZ compared to the one without VCZ in *Candida* sp.		[241]
PVA, SA (Sodium Alginate)	Electrospinning	Topic	242–542 nm	Cell viability of rat fibroblast cells was higher after crosslinking VCZ nanofibers.	There was a release of 38% of VCZ in 30 min and 89% in 8 h for the nanofibers. The kinetic model that explains the release was from Higuchi. Crosslinked or not with the nanofibers was more effective in promoting penetration into the skin layers than the control. After cross-linking, there was a reduction in the MIC value for *Candida* sp.		[242]
Compritol 888 ATO, Palmitic Acid, Stearic Acid, Glycerol, Soy Lecithin, Pluronic F-68, Sodium Tauracholate.	Emulsification Solvent Evaporation	Ocular	139–344 nm	In vitro studies of corneal hydration, histopathology and HET-CAM suggested a non-irritating property of the formulation.	Sustained release > 60% in 12 h of study.	Sustained release compared to VCZ suspension. A significantly lower amount of the drug was also observed in the plasma, suggesting nasolacrimal drainage.	[243]
Lecithin, Cholesterol	Film hydration	Ocular	*	Eye irritation studies in rabbits showed no irritation.	In vitro sustained release.		[244]
Soy Phosphatidylcholine, Cholesterol	Thin film hydration	Ocular	116 nm	HET-CAM irritability study indicated a non-irritating formulation, therefore, safe.	Mucin permeation study showed a good affinity with the mucous layer of the eye, showing ophthalmic viability.		[245]
Pluronic F-127 and F-68, Sodium Alginate	*In situ* gelation	Ocular	*		Antifungal activity in *C. albicans* and *A. fumigatus* depends on increased VCZ release from the gel. The formulation showed prolonged stability.		[246]
Chitosan, Silver, and Graphene Oxide	Electrostatic Interaction	Ocular	*	A study performed with corneal cells did not show cytotoxicity to the cells, demonstrating biocompatibility.	Sustained release of VCZ was observed through the hydrogel. The hydrogel showed inhibitory activity against *Fusarium solani* and *A. fumigatus* with MIC of 2.5 µg/mL and 2.5–5.0 µg/mL. The matrix activity of the contact lenses produced was also evaluated and presented a MIC of 1.25 µg/mL, suggesting an increase in therapeutic efficacy and a synergistic effect of the matrix with the hydrogel.	Clinical evaluation of rats treated with the lenses containing the VCZ hydrogel exhibited a reduction in fungal keratitis during the treatment period.	[247]
PLGA	Multiple Emulsion and Solvent Evaporation	Pulmonary	300 nm		In PBS medium, the drug release was limited by the dissolution rate of the drug particles, while in SLF medium, the release occurred by the diffusion/erosion mechanism.	There was variation in the concentration of VCZ in the pulmonary lobes of the animals during treatment with intravenous injection, a situation that did not occur with pulmonary administration. In addition, there was greater retention of VCZ in lung tissue from the nanoparticles.	[248]
Polietilenoimina, Stearic acid, sodium deoxycholate	Emulsification	Pulmonary	353 nm	Cytotoxicity in human lung carcinoma cells (A549) was dependent on polyethyleneimine concentration.	From the results of the mass mean aerodynamic diameter, the use of VCZ in the form of aerosols from agglomerates or nanoparticles can result in better pulmonary deposition than with the use of pure powder.		[159]
Carbopol 934, stearic acid, Tween 80	Ultrasonication and Microemulsion	Ocular	234–288 nm	The corneal hydration level remained between 76% and 79%, causing no damage to the corneal tissue.	Formulation using ultrasonication allowed controlled release for 12 h and prolonged stability. An Ex vivo transcorneal permeation showed controlled release in the cornea.		[46]
Jojoba Oil, Brij 97 and Sorbitol	Microemulsion	Topic	*		Microemulsion showed higher therapeutic efficacy than supersaturated VCZ solution against *C. albicans* ATCC 90028.		[249]
Precirol ATO 5, Labrafil 1944 CS, Tween 80	High Pressure Homogenization	Topic	210 nm		Sustained and controlled Release of VCZ for 24 h. The hydrogel formulation showed a higher amount of VCZ permeated in 12 h.		[160]

* Not mentioned.

## Data Availability

Not applicable.

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
