# Peer review of "Nanotechnology-Based Approaches for Voriconazole Delivery Applied to Invasive Fungal Infections"

_pharmaceutics, 2023, doi:10.3390/pharmaceutics15010266_

Round 1

Reviewer 1 Report

An interesting review is a study on "Nanotechnology-based approaches for voriconazole delivery applied to invasive fungal infections". However, the reader thinks that some issues should be corrected in order to complete and reconsideration. Therefore, the following items should be considered.

Comment 1:

Abstract section:

i.    Please rewrite the abstract. In the abstract, it is not very clear the purpose of the review is in accordance with the proposed title. Add some studies about nanotechnology-based approaches.

ii.  The authors also should include the main findings of the review in the abstract. The sentence "In this review, we highlighted recent works that have applied nanotechnology to deliver voriconazole more safely and effectively" only discusses the purpose of this review.

Comment 2:

Introduction section: The introduction section has been well structured and discussed in the manuscript.

Comment 3:

Line 395: Please look for other references regarding the range of nanoparticle sizes. I'm not sure about the sentence that the author wrote, " ....the application of particles with size in the range of 100-1000 nm (nanoparticles)...."

 Comment 4:

Table 2: Please add one column for the nanoparticle size used. This will further highlight the support in the nanotechnology aspect.

Comment 5:

Point 4 (4.1: 4.2: 4.3): This manuscript is an articles review regarding nanotechnology-based voriconazole delivery systems. However, the core section is lacking in the discussion. Please add some discussion from several references in this section.

Comment 6:

If possible, add a Future Prospect section for nanotechnology-based voriconazole delivery systems. This will add value to the importance of this article review.

Author Response

Thank you very much for your careful reading and comments about the manuscript.

Comment 1:

Abstract section:

  1. Please rewrite the abstract. In the abstract, it is not very clear the purpose of the review is in accordance with the proposed title. Add some studies about nanotechnology-based approaches.
  2. The authors also should include the main findings of the review in the abstract. The sentence "In this review, we highlighted recent works that have applied nanotechnology to deliver voriconazole more safely and effectively" only discusses the purpose of this review.

Answer: We appreciate the excellent observation. We improved the abstract section and add the main findings in the abstract.

In abstract: “Invasive fungal infections increase mortality and morbidity rates worldwide. The treatment of these infections is still limited due to low bioavailability and toxicity, requiring therapeutic monitoring, especially in the most severe cases. Voriconazole is an azole widely used to treat invasive aspergillosis, other hyaline molds, many dematiaceous molds, Candida spp, including those resistant to fluconazole, and for infections caused by endemic mycoses, in addition to those that occur in the central nervous system. However, despite its broad activity, using voriconazole has limitations related to its non-linear pharmacokinetics, leading to supratherapeutic doses and increased toxicity according to individual polymorphisms during its metabolism. In this sense, nanotechnology-based drug delivery systems have successfully improved the physicochemical and biological aspects of different classes of drugs, including antifungals. In this review, we highlighted recent work that has applied nanotechnology to deliver voriconazole more safely and effectively. These systems allowed increased permeation and deposition of voriconazole in target tissues from a controlled and sustained release in different routes of administration as ocular, pulmonary, oral, topical, and parenteral. Thus, nanotechnology application aiming to delivery voriconazole becomes a more effective and safer therapeutic alternative in the treatment of fungal infections”.

Comment 2:

Introduction section: The introduction section has been well structured and discussed in the manuscript.

Answer: Thank you very much for your comments.

Comment 3:

Line 395: Please look for other references regarding the range of nanoparticle sizes. I'm not sure about the sentence that the author wrote, " ....the application of particles with size in the range of 100-1000 nm (nanoparticles)...."

Answer: As there are disagreements between different authors about the size range of nanoparticles, we chose to modify the text and keep it more comprehensive.

Comment 4:

Table 2: Please add one column for the nanoparticle size used. This will further highlight the support in the nanotechnology aspect.

Answer: We have added a column containing the size range of nanoparticles, exemplifying authors who did not demonstrate the information in the text.

Comment 5:

Point 4 (4.1: 4.2: 4.3): This manuscript is an articles review regarding nanotechnology-based voriconazole delivery systems. However, the core section is lacking in the discussion. Please add some discussion from several references in this section.

Answer: We added more information in the text as suggested on the nanoparticles described in sections 4.1, 4.2, and 4.3, seeking to improve the discussion about the different systems.

In 4.1: “In general, lipid nanoparticles containing voriconazole have been developed mainly for ocular delivery this is justified by the compatibility and permeation capacity of these systems in this tissue, favoring a safe delivery of drugs. Studies with SLNs have shown increased permeation, mainly ocular, and allowed controlled and sustained release, increasing the therapeutic efficacy of these systems for VCZ delivery [46,245,246]”.

In 4. 2: “The characteristic of allowing the incorporation of different molecules into the polymeric nanoparticle allows for an increase in the interaction of the nanostructured system with the target tissue [231,285,286]. This has been explored in the literature with chitosan, which is known to have a mucoadhesive property and allows to increase in the delivery of VCZ in the target tissue [220,223,232,249]”

In 4.3: “Actually, nanoparticles to delivery voriconazole it promising their application varies according to the need for the route of administration. As this study, it is possible to see that lipid formulations seem to improve delivery by the ocular and topical route and protein and polymeric formulations were explored for oral and pulmonary delivery [219,224,236,239,242]. The physicochemical characteristics of nanoparticles are important to treat fungal infections, since each fungus develops better in a specific site”. “In general, using nanotechnology for voriconazole encapsulation and release is a strategy to increase therapeutic activity since studies have reported increased tissue permeation with reduced toxicity, mainly due to the observed sustained and controlled release [213,218]. In addition, it is possible to observe an improvement in the solubility and bioavailability of VCZ [232,239]”.

Comment 6:

If possible, add a Future Prospect section for nanotechnology-based voriconazole delivery systems. This will add value to the importance of this article review.

Answer: Thank you for your comment. We have added a new section about Future prospect, which greatly adds to the purpose of the article.

In 7. Future prospect: “It was observed that advances in the production of nanostructures containing voriconazole with a focus on topical, ocular and pulmonary administration are significant in relation to other drug application routes. Despite the diversity of developed systems, there are still limitations regarding efficacy and safety studies. Most of the studies compiled in this work employ in vitro tests to demonstrate its performance, while in vivo tests are little explored. The reduced applicability of in vivo tests and the lack of use of alternative models, both used for toxicity and pharmacokinetic assessments, indicate the need for efforts in this area. Another point is the development of new drugs using nanotechnology, where clinical studies are rare, which indicates that there is still much to be done for research to get off the ground and benefit the population. In this sense, the application of nanotechnology is encouraged for better performance of fungal drugs, which allow more efficacy and safety for the treatment of different fungal infections that explore the interaction of nanoparticles with yeast and filamentous fungi, improving the perspective of application of these nanoparticles”.

Reviewer 2 Report

The paper by Campos et. al. is well written and provides a god overview of nanoparticle-based technologies for voriconazole (VCZ) delivery. Overall, the information is presented in a easy to follow manner. Some minor suggestions are as follows:

Abstract: include info on why/how nanotechnology is essential in this case

Section 2: Recommend to add PK aspects in each sub-section, to showcase the need for controlled delivery

Section 2: This section could be shorter, as it's not the main focus of the article

Section 3: Add a physicochemical section for VCZ

Section 6: This section should be expanded to include more info on ongoing clinical trials that utilize nanotechnology to deliver VCZ.

Author Response

Thank you very much for your careful reading and comments about the manuscript.

Abstract: include info on why/how nanotechnology is essential in this case

Answer: The abstract section has been rewritten to address the questions raised by the reviewer about why and how nanotechnology is important in this case.

In abstract: “Invasive fungal infections increase mortality and morbidity rates worldwide. The treatment of these infections is still limited due to low bioavailability and toxicity, requiring therapeutic monitoring, especially in the most severe cases. Voriconazole is an azole widely used to treat invasive aspergillosis, other hyaline molds, many dematiaceous molds, Candida spp, including those resistant to fluconazole, and for infections caused by endemic mycoses, in addition to those that occur in the central nervous system. However, despite its broad activity, using voriconazole has limitations related to its non-linear pharmacokinetics, leading to supratherapeutic doses and increased toxicity according to individual polymorphisms during its metabolism. In this sense, nanotechnology-based drug delivery systems have successfully improved the physicochemical and biological aspects of different classes of drugs, including antifungals. In this review, we highlighted recent work that has applied nanotechnology to deliver voriconazole more safely and effectively. These systems allowed increased permeation and deposition of voriconazole in target tissues from a controlled and sustained release in different routes of administration as ocular, pulmonary, oral, topical, and parenteral. Thus, nanotechnology application aiming to delivery voriconazole becomes a more effective and safer therapeutic alternative in the treatment of fungal infections”.

Section 2: Recommend to add PK aspects in each sub-section, to showcase the need for controlled delivery.

Answer: Thank you for your comment, but the purpose of this section was to address invasive fungal pathologies. Where information on voriconazole PK was covered in the specific session 3.0.

Section 2: This section could be shorter, as it's not the main focus of the article.

Answer: Thank you for your comment, but we chose to keep the session the size it is because of the recommendations of the other reviewers.

Section 3: Add a physicochemical section for VCZ

Answer: Thank you for your suggestion, this information is very important and was missing from the initial text but has been added.

Line 345/346: It has low solubility in water (0,098 mg/mL), log P 1,82, and pKa 2,01 and 12,7 [136].

Section 6: This section should be expanded to include more info on ongoing clinical trials that utilize nanotechnology to deliver VCZ.

Answer: A new survey was carried out and we identified that new studies with voriconazole were registered on the Clinical Trials website. However, all studies were reviewed and only two studies that evaluated voriconazole nanoparticles were included. Thus, we complement the text with more information about the studies.

Reviewer 3 Report

The review focusing on developing nanoparticle-based formulations for voriconazole is well written. However, the manuscript needs to be updated based on the below comments. 

  1. The abstract needs to be improved. It is written very generally. 
  2. Among various fungal infections, which are the most common, and what percentage of the population is usually affected? Can authors please make a note of it? 
  3. Please provide additional information related to the drug substance (voriconazole), such as solubility (pH-dependent solubility if available), log P, pKa, and any stability issues if exist. 
  4.  Please make a note of the commercial formulations. 
  5.  Among various approaches for developing nanoparticles, please provide additional information discussing which approach is superior and when to choose it.   

Author Response

Thank you very much for your careful reading and comments about the manuscript.

  1. The abstract needs to be improved. It is written very generally.

Answer: The abstract section has been rewritten more extensive:

In abstract: “Invasive fungal infections increase mortality and morbidity rates worldwide. The treatment of these infections is still limited due to low bioavailability and toxicity, requiring therapeutic monitoring, especially in the most severe cases. Voriconazole is an azole widely used to treat invasive aspergillosis, other hyaline molds, many dematiaceous molds, Candida spp, including those resistant to fluconazole, and for infections caused by endemic mycoses, in addition to those that occur in the central nervous system. However, despite its broad activity, using voriconazole has limitations related to its non-linear pharmacokinetics, leading to supratherapeutic doses and increased toxicity according to individual polymorphisms during its metabolism. In this sense, nanotechnology-based drug delivery systems have successfully improved the physicochemical and biological aspects of different classes of drugs, including antifungals. In this review, we highlighted recent work that has applied nanotechnology to deliver voriconazole more safely and effectively. These systems allowed increased permeation and deposition of voriconazole in target tissues from a controlled and sustained release in different routes of administration as ocular, pulmonary, oral, topical, and parenteral. Thus, nanotechnology application aiming to delivery voriconazole becomes a more effective and safer therapeutic alternative in the treatment of fungal infections”.

  1. Among various fungal infections, which are the most common, and what percentage of the population is usually affected? Can authors please make a note of it?

Answer: We highlight information from the previous paper on incidence rates and include new data in the section:

In 2.1.3 invasive aspergillosis: “The global prevalence of aspergillosis reaches an estimated 3,000,000 cases per year of chronic pulmonary aspergillosis and 300,000 cases per year of IA [75] “.

In 2.2 Candida infections: “There are more than 15 species of Candida sp capable of causing infections in humans; however, the five most common species that cause infections are C. albicans, C. glabrata, C. tropicalis, C. parapsilosis, and C. krusei, being C. albicans responsible for 40 to 60% of cases worldwide [91–94]”.

In 2.3 Cryptococcosis: “Cryptococcosis was recently highlighted by the world health organization as the first highest-priority microorganism, surpassing even Candida C. auris, more recently described and of worldwide concern due to its resistance. Inserting new therapies for mycoses, especially those pointed out here, is essential in this context, according to the number of deaths from HIV associated cryptococcol meningits , an estimated 181,000 cases worldwide represent 15% of all AIDS-related deaths [1,21] .

In addittion, we reinforce in the introduction that the priority criterion for fungal infections is based on multicriteria and not just on incidence and prevalence, hence the limitation in estimating the data.

  1. Please provide additional information related to the drug substance (voriconazole), such as solubility (pH-dependent solubility if available), log P, pKa, and any stability issues if exist.

Answer: We added the information related to voriconazole:

In 3.1 general aspects: “VCZ (Vfendâ by Pfizerâ, tablet and IV) is an antifungal belonging to the azole class, derived from a structural modification of fluconazole and approved by the Food and Drug Administration (FDA) in 2002 [134,135]. It has low solubility in water (0,098 mg/mL), log P 1,82, and pKa 2,01 and 12,7 [136].”.

  1. Please make a note of the commercial formulations.

Answer: Commercial formulations were included:

In 3.1 general aspects: “VCZ (Vfend by Pfizer) is an antifungal belonging to the azole class, derived from a structural modification of fluconazole and approved by the Food and Drug Administration (FDA) in 2002[134,135].

  1. Among various approaches for developing nanoparticles, please provide additional information discussing which approach is superior and when to choose it.

Answer: We included a discussion on which system was most suitable for the route of administration of interest for voriconazole delivery...

In 4.1: “In general, lipid nanoparticles containing voriconazole have been developed mainly for ocular delivery this is justified by the compatibility and permeation capacity of these systems in this tissue, favoring a safe delivery of drugs. Studies with SLNs have shown increased permeation, mainly ocular, and allowed controlled and sustained release, increasing the therapeutic efficacy of these systems for VCZ delivery [46,245,246]”.

In 4. 2: “The characteristic of allowing the incorporation of different molecules into the polymeric nanoparticle allows for an increase in the interaction of the nanostructured system with the target tissue [231,285,286]. This has been explored in the literature with chitosan, which is known to have a mucoadhesive property and allows to increase in the delivery of VCZ in the target tissue [220,223,232,249]”

In 4.3: “Actually, nanoparticles to delivery voriconazole it promising their application varies according to the need for the route of administration. As this study, it is possible to see that lipid formulations seem to improve delivery by the ocular and topical route and protein and polymeric formulations were explored for oral and pulmonary delivery [219,224,236,239,242]. The physicochemical characteristics of nanoparticles are important to treat fungal infections, since each fungus develops better in a specific site”. “In general, using nanotechnology for voriconazole encapsulation and release is a strategy to increase therapeutic activity since studies have reported increased tissue permeation with reduced toxicity, mainly due to the observed sustained and controlled release [213,218]. In addition, it is possible to observe an improvement in the solubility and bioavailability of VCZ [232,239]”.